# Mobile phones and infant health at birth

Luca Maria Pesando[1,2,3]*, Komin Qiyomiddin[2]

**1** Division of Social Science, New York University (AD), Abu Dhabi, UAE, **2** Department of Sociology, School of Arts, McGill University, Montreal, Canada, **3** Weatherhead Center for International Affairs, Harvard University, Cambridge, MA, United States of America

* lucamaria.pesando@nyu.edu

## Abstract

There is increasing evidence that digital technologies such as mobile phones have the potential to shape some of the United Nations (UN) Sustainable Development Goals (SDGs) such as health, education, and nutrition, even among the most resource-deprived countries and communities in the world. Nonetheless, little research has focused on the intergenerational implications of digital technologies for infant health and wellbeing. This study leverages Demographic and Health Survey (DHS) data from 29 low- and middle-income countries (LMICs) to explore associations between mothers' ownership of mobile phones and their children's health at birth, as measured by birth weight and low birth weight (LBW), i.e., weight lower than 2,500 grams. Infants born to women owning mobile phones fare consistently better in terms of birth weight, even after accounting for potential socioeconomic confounders and other sources of media or information in the household. Partly, mechanisms are consistent with the idea of broader knowledge and access to healthcare services, as associations are mediated by a higher number of antenatal visits, higher likelihood of having a birth assisted by a health professional, and by the extent to which mothers hear about family planning by text message. Associations are strongest among low-educated mothers. Also, associations are stronger in countries where infant health is poorer yet mobile-phone diffusion is higher, highlighting the comparatively higher potential of the diffusion of mobile phones for global development in poorest contexts. Our findings may be of interest to scholars and policymakers concerned with identifying relatively cheap policy levers to promote global health and wellbeing in disadvantaged contexts, particularly among women.

## Introduction

Across low- and middle-income countries (LMICs), Information and Communication Technologies (ICTs) hold huge promise for sustainable development, poverty reduction, and gender equality. For instance, Target 5B of the United Nations (UN) Sustainable Development Goal (SDG) 5 indicates the enhanced use of "enabling technology" as a pathway through which to promote women's empowerment [1]. Similarly, through its global "digital health" agenda, the World Health Organization (WHO) is harnessing the power of ICTs and health innovations to accelerate global attainment of health and wellbeing [2]. Among these ICTs,

**Data Availability Statement:** The dataset used in the current analysis can be obtained from DHS Program: https://dhsprogram.com/. Any researcher can access the data from DHS databases after registering at: https://dhsprogram.com/data/new-user-registration.cfm and get

permission to access data at https://dhsprogram.com/data/available-datasets.cfm. This study covers 29 countries and all countries and survey waves are listed in Supporting Information Table A1.

**Funding:** Pesando acknowledges financial support from the Social Sciences and Humanities Research Council of Canada (SSHRC), Insight Development Grant # 430-2021-00147 (PI: Pesando), the Jacobs Foundation Fellowship (grant # 2021-1417-00) from the Jacobs Foundation, and the Division of Social Science at New York University (NYU) Abu Dhabi (grant # 76-71240-ADHPG-AD405).

**Competing interests:** The authors have declared that no competing interests exist.

due to their rapid diffusion–which in many LMICs has fully outpaced the "landline phase"– mobile phones have increasingly attracted the attention of scholars and policymakers as rather inexpensive devices that can be leveraged to promote sustainable development and, ultimately, attain a range of development outcomes that go beyond gender equality (SDG5), such as better education (SDG4), higher economic growth (SDG8), reduced poverty (SDG1) and–as relevant to this study–improved health and wellbeing (SDG3) [3–6]. Some of the latest cross-national research on the topic suggests that, especially when it comes to women's reproductive health, mobile phones have become a key tool for empowering women and bringing about behavioral changes in LMICs [7], while the role of television and radio have followed opposite trajectories, i.e., their importance has diminished significantly over time [8].

Despite the growing body of social science research highlighting the potential of mobile phones to advance global social development, little research to date has focused on the *intergenerational* implications of digital technologies for child health and wellbeing. This focus is important for at least two reasons. First, as infant health is instrumental for a whole range of later-life outcomes including educational attainment and later-life health [9], identifying a potential driver of infant health may expand our understanding of broader life-course dynamics. Second, an explicit focus on child health at birth may more directly bring to light and reflect the extensive range of health-related functions that mobile phones enable for women during and after pregnancy, further discussed below.

While previous research has extensively discussed the capabilities of mobile phones, which have expanded from enabling communication and expanding community outreach to the provision of information and delivery of services–including health-related services–the health focus has primarily been on women's enhanced information and smoother access to sexual and reproductive health services, including maternal healthcare [10]. For instance, a recent study combining a wide range of data sources and methodologies suggests that women who own a mobile phone in resource-limited settings of the African region are better informed about sexual and reproductive health services and are better equipped to make independent decisions regarding contraception, with larger payoffs observed in more disadvantaged geographical areas [7]. At a macro-level scale (200+ countries), the same study also reveals that mobile-phone access is associated with lower gender inequality, higher contraceptive uptake, and lower maternal and child mortality.

Besides these macro-level associations, little micro-level evidence has focused on the extent to which mothers' ownership and use of mobile phones–by means of the virtuous channels outlined above–bears any relationship with their offspring health at birth. This is precisely the research question we set to address in the current study by leveraging micro-level data from Demographic and Health Surveys (DHS) from 29 LMICs and focusing on birth weight and low birth weight (LBW)–i.e., weight lower than 2,500 grams–as outcomes. The choice of birth weight as primary outcome rests on existing research suggesting that low birth weight is mostly a consequence of choices and constraints faced by mothers pre- and during pregnancy [11]. Relatedly, birth weight has considerable short- and long-term implications for individuals' later-life trajectories in both low-income and high-income societies. For instance, a twin study using the Early Childhood Longitudinal Study from the US reveals that lower-birth-weight children display worse cognitive and socioemotional outcomes prior to school entry (i.e., at age 4) [12]. Similarly, a twin study from Japan suggests that birth weight has a causal effect on academic achievement around the age of 15, yet it does not affect later-life earnings [13]. DHS data are valuable in this respect due to their broad country coverage, adequate sample sizes, and detailed information on child health that can be linked to mothers, alongside relatively recent information on individual-level ownership of mobile phones, as well as household-level information on other media sources.

## Evidence from LMICs on the role of mobile phones in the health arena and contribution of this study

Research on the role of mobile phones in promoting better health across LMICs has been growing gradually over the last decade, partly in response to the implementation of a series of technological interventions [1, 3]. Relatedly, health practitioners have been at the forefront of using mobile phones as a development tool to implement "mobile-health" (*mHealth*) projects. *MHealth* projects range in variety and scope, from monitoring measles outbreaks in Zambia, to supporting diagnosis and treatment by health workers in Mozambique, to sending health education messages in Benin, Malawi, and Uganda [3]. In other countries such as Kenya, Malaysia, and South Africa, mobile phones have been used to send reminders to HIV-positive patients about their anti-retroviral therapy (ART) schedule, as well as to allow community health workers to send information about HIV patients' status [1, 14, 15].

Mobile phones are also currently used among women to monitor their children's health, both pre- and post-delivery. For instance, in the Democratic Republic of Congo mothers can use mobile phones to call a helpline to inquire and ask questions about their child's health status [16]. As elaborated in previous work [1], a specific focus on women's sexual and reproductive health shows effects of *mHealth* interventions on improved antenatal care attendance [17, 18], reduced perinatal mortality [19], improved clinical outcomes of HIV-positive pregnant women [20], higher contraceptive use [21, 22], increased contraceptive acceptability [23], and higher vaccination uptake [24] in contexts as diverse as Bangladesh, Nigeria, Palestine, South Africa, Zanzibar, etc. While not specifically focused on *mHealth* interventions, a recent study also explored the relationship between mobile-phone ownership and health behaviors of post-partum mothers in the rural Ntcheu district of Malawi, finding robust positive associations with exclusive breastfeeding practices and, to a lesser extent, fewer depressive symptoms and higher social support post-partum [10].

Concrete and successful examples of *mHealth* interventions targeting women were promoted by the South African National Department of Health starting in 2014 [25]. *MomConnect* is a notable example, itself designed on the examples of *Aponjon* in Bangladesh, *Wazazi Nipendeni* in Tanzania, and *Chipatala Cha Pa Foni* in Malawi. *MomConnect* is an online registration system that enrolls each pregnant woman in South Africa into a pregnancy database complemented with a text-based platform that sends weekly messages with information on how to carry out a healthy pregnancy and access newborn care through a help desk that allows women to ask any additional question they may have, as well as voice any worry or concern [26]. These interventions well exemplify the potential role that mobile phones may play when it comes to influencing infant health by means of improved knowledge, which in turn reflects onto better maternal health. More details on the topic can be found in a review study focused on the role of mobile phones and gender-related outcomes [1].

On the other hand, country-specific studies have highlighted important contextual factors and barriers that may prevent some women from benefitting fully from digital-technology interventions [1]. For instance, researchers have investigated the usage of mobile phones for seeking childbirth services in Bangladesh and have documented a low rate of utilization among urban women, with important variation among slum and non-slum dwellers, mostly due to socioeconomic, neighborhood, and environmental barriers, alongside poor literacy skills (skill-related digital divides) [19]. Other scholars also acknowledged the multiple opportunities that mobile phones may open up for pregnant women in Malawi and Nigeria in terms of lower maternal morbidity and mortality and enhanced informational, economic, and psychological well-being [18, 27]. Yet they also stress that the realization of outcomes is mediated by multiple intertwined personal, social, and environmental factors that can serve as barriers,

such as unequal gender dynamics, negative community attitudes, lack of family support, poor infrastructure, cost of data plans, etc.–highlighting again access (*first-level*) and skill-related (*second-level*) digital divides [1, 18, 27].

Despite these barriers, which call for policies to bridge digital divides by gender and across rural and urban areas–as well as interventions aimed at lowering the cost of data plans–mobile phones and *mHealth* interventions enable a myriad of opportunities in the health arena, especially for women. While the existing literature has primarily focused on women's reproductive health before and during pregnancy, no study has focused explicitly on the relationship between mothers' mobile-phone ownership and infants' health as measured by birth weight and LBW as we do in this study. This is an important omission, as it would be reasonable to expect the ones that have been identified as outcomes in previous research–such as improved antenatal care attendance [10, 17, 18] and improved health-related knowledge through interventions such as *MomConnect* [25, 26]–to serve as key channels that may lead to improved infant health at birth. As such, given that birth weight reflects, among other non-modifiable characteristics such as height, age, and parity, choices, behaviors, and constraints faced by mothers pre- and during pregnancy [11], and it has been shown to be one of the strongest predictors of later-life educational and health outcomes [12, 13, 28], the focus on birth weight is all the more appropriate.

Based on the existing research, we envision three main channels through which mobile-phone ownership may be related to infant health at birth, namely: 1) expanded *access to health-care services* in the prenatal and postnatal periods, including services that can be booked, accessed, and executed remotely; 2) broader *knowledge and awareness* of reproductive-health issues, including improved access to health-related information (e.g., receiving regular text messages with information on how to carry out a healthy pregnancy as done, for instance, through *MomConnect*); and 3) wider *communication* and enlarged social networks, including with family members and other women in the community with whom to exchange information or share health concerns.

## Materials and methods

This paper uses data from the Demographic and Health Survey (DHS) program. We utilize the latest DHS surveys that include a specific question asking respondents–women, in this case– about individual-level mobile-phone ownership. Initially, we assembled 31 countries' latest DHS surveys conducted between 2015 and 2020 and containing a question about women's mobile-phone ownership. We then dropped any country with a prevalence of mobile-phone ownership of 95% or above to allow for adequate variation in the main predictor of interest. After a quick calculation determining the share of women owning mobile phones in each country, Armenia and Maldives were excluded from the final sample selection as their percentages were above 95% (see S1 Table in S1 File for details on countries and waves). The combined sample includes 170,916 women (mothers) aged 15–49 from 29 countries–in alphabetical order, Albania, Angola, Bangladesh, Benin, Burundi, Cameroon, Ethiopia, Gambia, Guinea, Haiti, Indonesia, Jordan, Liberia, Mali, Malawi, Nigeria, Nepal, Philippines, Pakistan, Rwanda, Sierra Leone, Senegal, Tajikistan, Timor Leste, Tanzania, Uganda, South Africa, Zambia, and Zimbabwe.

We restricted the sample to mothers who gave birth in the three years preceding the survey–with robustness checks provided on 1) the sample restricted to mothers who gave birth over the preceding year (sample size smaller by about a half), as well as 2) the sample limited to the most recent birth for each mother–both reported in the S1 File. We chose to keep all births occurring over the past three years to retain an adequate sample size to obtain accurate

estimates, especially for conducting country-specific analyses. We decided to keep relatively "recent" births to account for potential temporal-ordering concerns whereby some births may precede mobile-phone ownership, thus invalidating our logic. As a matter of fact, the mobile-phone variable in the survey asks a simple question about whether the respondent owns a mobile phone vs not. As such, the DHS survey contains no information in relation to the timing of mobile-phone purchase, which may lead to temporal-ordering–and, in turn, reverse-causality–concerns. By restricting to these births, we are hoping to minimize such concern. Results over the three-year sample and the one-year sample provide virtually identical results.

From a methodological standpoint, we begin by providing graphical descriptive evidence of the relationship between mobile-phone ownership and whether the child is considered a LBW infant. The cut-off selected for low-weight births is when a child weighs 2,500 grams or less [28–31], a variable which was created from the original continuous one, also kept as a separate independent outcome–as continuous and dichotomous outcomes may retain different amounts of information. As far as the outcomes are concerned, three observations are worth of note. First, while the sample is comprised of approximately 170,000 mothers, about 7% of them do not recall the weight of their baby at birth, while for 30% of them their babies were not weighed whatsoever, leaving us with an actual analytical sample of 108,103 mothers with complete information on birth weight. Attrition analyses reported in S2 Table in S1 File suggest that, as expected, women with incomplete birth-weight information–i.e., missing for whatever reason–are significantly less likely to own a mobile phone, they are younger, less educated, poorer, and more likely to live in rural areas relative to women with complete birth-weight information–a series of aspects suggesting that we are dealing with a selected sample of "advantaged" mothers. This finding should be kept in mind when thinking about external validity and will be further addressed in the remainder of the study. Second, while we restrict the sample to births occurring over the previous three years, some mothers may have had multiple births over such timeframe. In fact, about 87% of women had one birth, 12% two births, and 0.42% three or more births. As such, our database is not at the woman level, but at the woman-child level, delivering a database of 152,172 observations with complete mobile-phone ownership and birth-weight information. In other words, over the previous three years, 170,916 mothers had 260,158 births, of which only 152,172 have complete birth-weight information. Third, the average birth weight on the pooled analytical sample is around 3,168 grams and about 16.5% of all births are classified as LBW. Again, if we had complete birth-weight information on all women, the share of LBW births would likely be significantly higher.

Descriptive graphs are provided for all countries combined, as well as separately by country. We then run a series of ordinary least-squares (OLS) regressions predicting infants' birth weight both as continuous and categorical variable (LBW)–with robustness checks with non-parametric matching techniques provided in the S1 File–and assess heterogeneity in the estimates by individual- and country-level characteristics. Among the individual-level characteristics, we focus on women's level of education and household location of residence (rural/urban). Among the country-level characteristics, we focus on four proxies of infant health, namely prevalence of LBW (obtained from UNICEF), prevalence of stunting, wasting, and underweight, the latter three obtained from the DHS StatCompiler (hence, including the whole sample of women, not only "recent" mothers).

For the main analyses, we estimate three models, with sociodemographic and socioeconomic control variables added sequentially. Each model is estimated separately for the continuous outcome and the dichotomous one. The first model provides a basic bivariate association; individual-level characteristics are added to model 2, namely respondent's age, marital status with a total of six categories (never in a union, married, living with a partner, widowed, divorced, no longer living together/separated), currently working (yes/no),

education (none, primary, secondary, and higher), and sex of the child; model 3 ("full specification") adds household-level characteristics, which include the wealth index in categories (poorest, poorer, middle, richer, and richest), location of residence (urban/rural), and two dummy variables indicating whether the household has television and radio (yes/no). Lastly, in separate analyses we complement the full specification by exploring potential channels or mediators through which mobile-phone ownership may relate to infants' birth weight. These channels mirror the ones discussed in the above section, namely better access to healthcare services in the prenatal and postnatal periods, broader access/knowledge of health-related information, and expanded communication. We measure the first channel using proxy variables such as the number of antenatal visits (prenatal period) and whether the actual birth was assisted by professional staff (postnatal period). Similarly, we measure the second channel using variables that proxy for respondents' knowledge of health-related behavior (not exclusively tied to pregnancy), such as whether they possess knowledge of a place to get HIV-tested, knowledge of any contraceptive method, and whether they received any information about family planning by text message. As for the third channel, the DHS does not provide suitable information on communication channels or social networks, hence we will mostly speculate on its role in the current study. Analyses of infants' birth weight outcomes are weighted using the appropriate weights provided by the DHS and account for the complex DHS survey design by adjusting the standard errors for cluster sampling at the level of the primary sampling unit (PSU).

In addition, although not shown in the main results, all regressions control for country dummy variables and year of the survey, and–for country-specific analyses–interactions between country dummies and mobile-phone ownership. We control for the variable country because our data belong to different countries and each survey has been administered in different years for different countries. Furthermore, from specification 2 onwards we also control for child's birth order–also omitted from the tables. The inclusion of child's birth order enables us to account for the possibility that some women may have had more than one child over the three years preceding the survey and that infant health may differ across parities. In fact, there is evidence that infant health outcomes may be largely different between first and subsequent births, as the aetiology of births varies importantly by parity, especially between first and subsequent births [32].

Table 1 provides summary statistics on the analytical sample of women (mothers) and countries, for all 29 countries combined. Across all countries, about 62% of women own a mobile phone. Mothers are, on average, 28 years old; 75% of them are married, 55% are currently working and almost 50% of them have some form of secondary education or more. The majority (about 56%) of households are rural and 52% and 46% own a TV and a radio, respectively. On average, mothers had five antenatal visits before giving birth, and only 26% of them had birth assisted by a health professional. Moving to countries, the average prevalence of LBW and stunting are 15% and 30%, respectively, and 59% of women own a mobile phone.

## Descriptive evidence

Fig 1 provides the distribution of the main predictor (top panel) and outcome (bottom panel) of interest among our analytical sample of mothers, sorted by country. Both variables show a high degree of cross-country variability. In terms of mobile-phone ownership, Ethiopia features at the bottom of the distribution (18%), closely followed by Burundi (21%) and Malawi (27%), while South Africa (90%), Jordan (92%), and Albania (92%) feature at the top. In terms of LBW, in Pakistan as high as 36% of births are classified as LBW, followed by Bangladesh (31%) and Nepal (29%), while infant health is best in and Albania, with 8% of births classified as LBW, followed by Sierra Leone (9.5%) and Tajikistan (10%).

**Table 1. Descriptive statistics on the analytical sample of women and countries.**

| | Mean, prop. or % | (SD) | Min. | Max. |
|---|---|---|---|---|
| *Women (analytical sample of mothers)* | | | | |
| Mobile-phone ownership | 0.620 | (0.485) | 0 | 1 |
| Age | 28.40 | (6.647) | 15 | 49 |
| Marital status | | | | |
| Never in union | 6.540 | | | |
| Married | 75.12 | | | |
| Living with partner | 13.27 | | | |
| Widowed | 0.77 | | | |
| Divorced | 1.60 | | | |
| No longer living together | 2.71 | | | |
| Currently working | 0.549 | (0.498) | 0 | 1 |
| Education | | | | |
| None | 19.92 | | | |
| Primary | 30.99 | | | |
| Secondary | 37.88 | | | |
| Higher | 11.21 | | | |
| Wealth index | | | | |
| Poorest | 16.9 | | | |
| Poorer | 19.01 | | | |
| Middle | 20.29 | | | |
| Richer | 21.6 | | | |
| Richest | 22.21 | | | |
| Urban residence | 0.437 | (0.496) | 0 | 1 |
| Owns TV | 0.521 | (0.500) | 0 | 1 |
| Owns radio | 0.460 | (0.498) | 0 | 1 |
| Knows place to get HIV tested | 0.846 | (0.361) | 0 | 1 |
| Knows modern method of contraception | 0.985 | (0.120) | 0 | 1 |
| Heard family planning by text on phone | 0.057 | (0.233) | 0 | 1 |
| Number of antenatal visits | 5.242 | (2.991) | 0 | 36 |
| Birth assisted by health professional | 0.255 | (0.425) | 0 | 1 |
| *Countries* | | | | |
| Prevalence of LBW (UNICEF) | 14.94 | (5.311) | 4.60 | 34.05 |
| Stunting prevalence (DHS) | 30.15 | (9.807) | 7.70 | 55.90 |
| Wasting prevalence (DHS) | 6.131 | (4.343) | 1.10 | 24.0 |
| Underweight prevalence (DHS) | 15.42 | (8.366) | 1.50 | 40.40 |
| Women owning mobile phone, % (DHS) | 59.39 | (18.31) | 23.60 | 91.80 |
| Mobile-phone subscriptions per 100 people (WDI) | 88.22 | (30.95) | 36.13 | 164.5 |

*Notes*: SD = standard deviations (in parentheses). LBW = low birth weight. DHS = Demographic and Health Surveys. WDI = World Development Indicators. DHS sampling weights applied. LBW defined as weight below 2,500 grams.

Fig 2 illustrates the distribution of infants' birth weight (left) and LBW (right) among mothers with and without a mobile phone for all countries combined. The left panel reveals that there is virtually no difference in mean birth weight among these two groups of mothers. Conversely, the right panel suggests that 15.7% of low-birth-weight babies are born to mothers with mobile phones, whereas the estimate is around 17.7% for babies born to mothers without mobile phones. Although the evidence is purely descriptive–and with no controls accounted

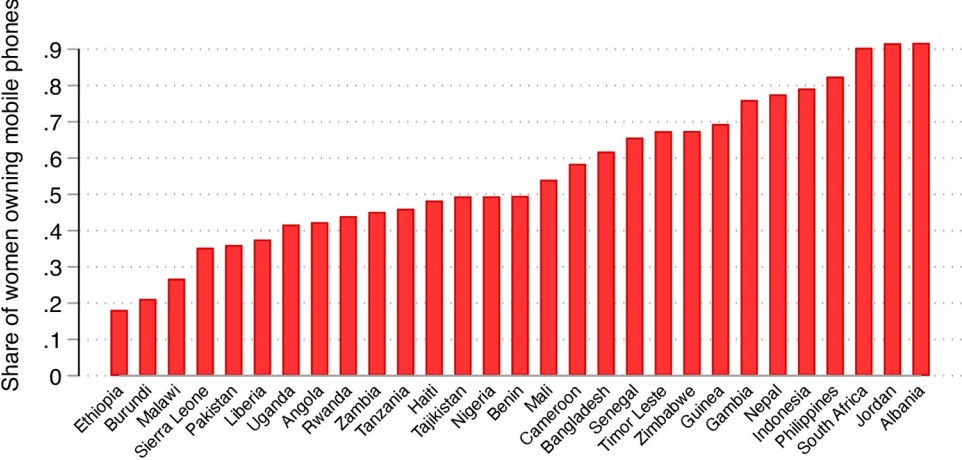

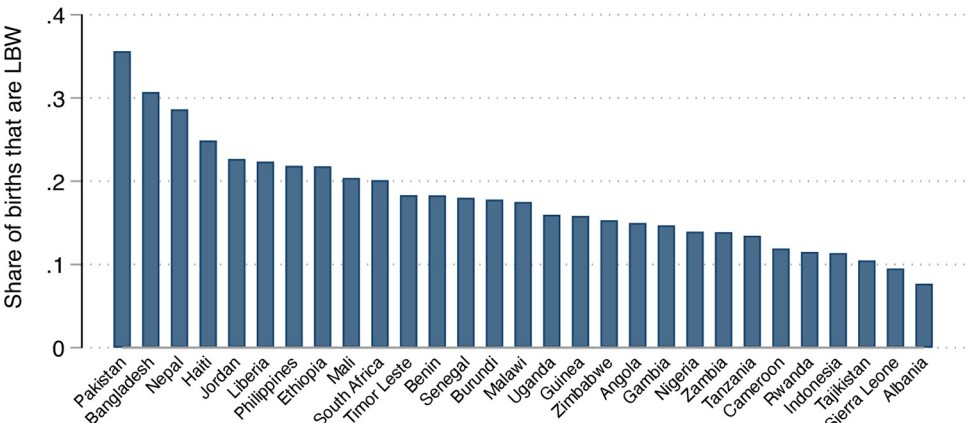

**Fig 1.** Distribution of mobile-phone ownership (top panel) and share of births that are LBW (bottom panel) by country, analytical sample of mothers. *Notes*: LBW = low birth weight. LBW defined as weight below 2,500 grams. DHS sampling weights applied.

for–summary statistics from this pooled sample suggest a non-negligible raw difference of 2 percentage points in newborns' health outcomes when mothers' mobile-phone ownership status is considered which, considering the mean LBW among women without mobile phones on the pooled sample (0.177), corresponds to an 11% lower LBW among women owning mobile phones. These descriptive findings–and, foremost, the difference between the two panels–also suggest that it is worth examining both health outcomes as they are not measuring fully overlapping constructs.

Fig 3 depicts the same relationship between mothers' mobile-phone ownership and the prevalence of LBW infants separately by country. The figure shows trends analogous to those presented in Fig 2 for all but three countries–Guinea, Liberia, and Sierra Leone–yet results are also very similar across the two groups (owning phones vs not) in Ethiopia, Haiti, Tanzania, and Zambia. Guinea, Liberia and Sierra Leone represent an "anomaly" as the prevalence of LBW babies is slightly higher among mothers with mobile phones, yet not significantly so. As the variability in LBW is high across countries (Fig 1, bottom panel) and the raw differences also vary widely across countries (Fig 3), percentage-point differences imply percentage

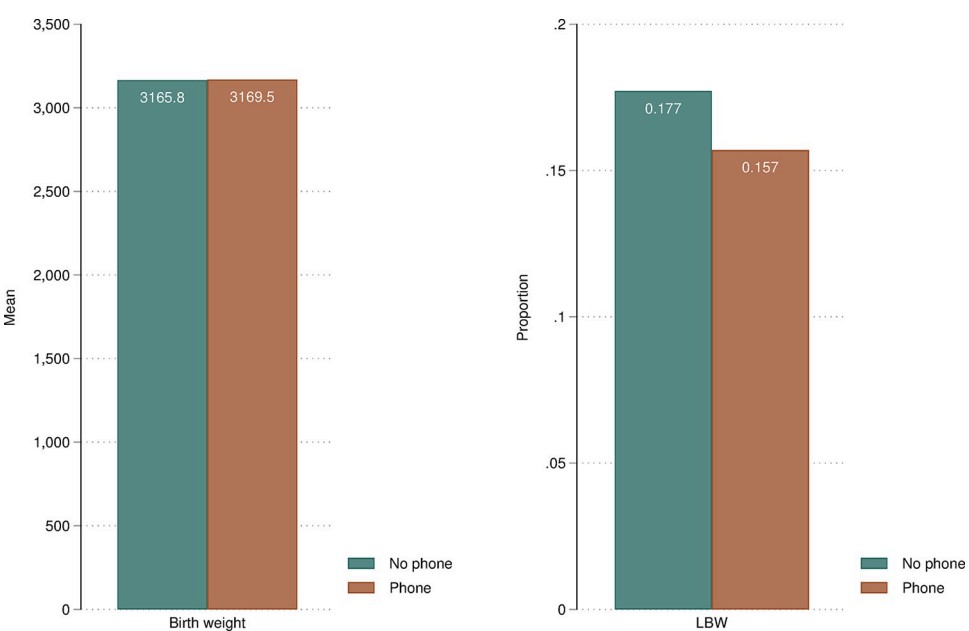

**Fig 2.** Distribution of birth weight (left panel) and LBW (right panel) by women's mobile-phone ownership status, analytical sample of mother-child. **Notes**: LBW = low birth weight. LBW defined as weight below 2,500 grams. DHS sampling weights applied.

changes ranging between 17% in Pakistan (the country with highest LBW) and 24% in Albania (the country with lowest LBW). These results are purely descriptive, hence the general pattern along with the few "anomalies" may be driven by a set of observable and unobservable

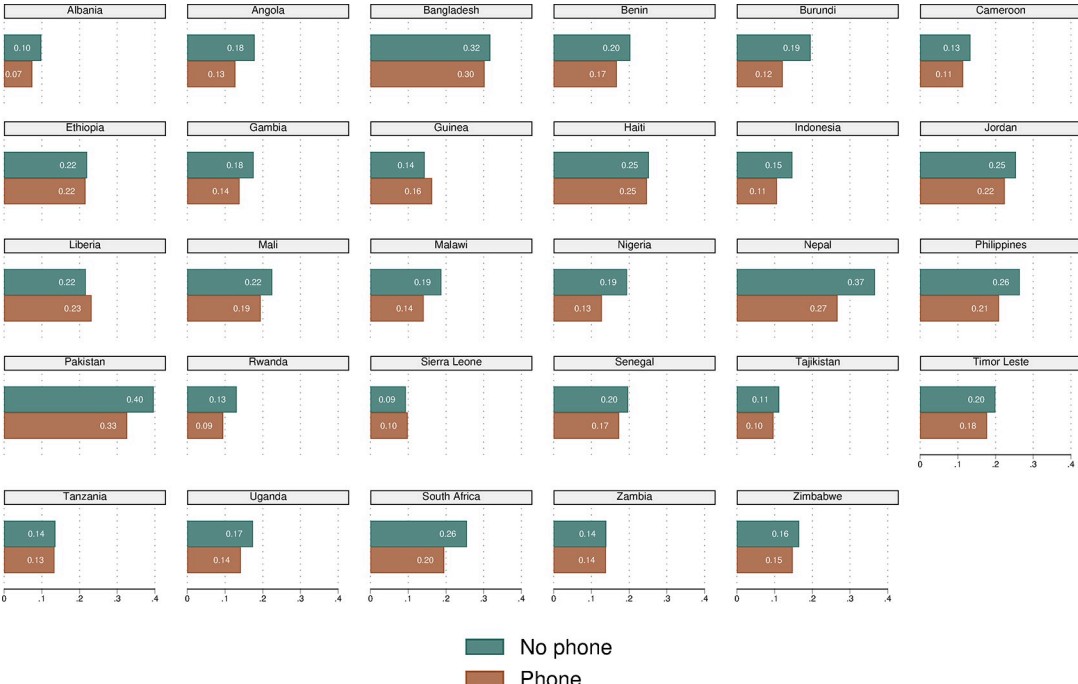

**Fig 3. Distribution of LBW by women's mobile-phone ownership status by country, analytical sample of mother-child.**
**Notes**: LBW = low birth weight. LBW defined as weight below 2,500 grams. DHS sampling weights applied.

characteristics at the individual and household levels. To account for more comprehensive analyses that include potential confounders, we thus turn to multivariate analyses.

## Results

### Individual-level characteristics

This section presents results from the main analyses. The estimated coefficients on the main predictor of interest across all three models are consistent with the descriptive evidence provided in Figs 2 and 3: the association between mothers' mobile-phone ownership and their newborns' health is positive. Specifically, Table 2 includes two columns per model, where the former ("continuous") presents estimated coefficients on mobile-phone ownership among mothers predicting birth weight as a continuous outcome, while the latter ("LBW") predicts the binary LBW outcome. Although all specifications control for country and survey-year dummies–as well as birth order of the child from specification 2 onwards–for the sake of conciseness these coefficient estimates are omitted from the tables.

Model 1 provides a bivariate association between mobile-phone ownership and newborn's birth weight accounting for country and year dummies. The estimated coefficient suggests that newborns whose mothers own mobile phones weigh on average 45.4 grams more relative to newborns whose mothers do not own mobile phones. Model 2, which includes individual-level controls (alongside birth-order dummies), also suggests that the estimated association is positive and statistically significant, yet the magnitude of the coefficient decreases from 45 to roughly 41, suggesting that individual-level characteristics contribute, albeit minimally, to explaining variability in birth weight. As for the controls, infants born to ever-married mothers (be them still married, widowed, or divorced) are more likely to be healthier compared to newborns born to never-married mothers. Mothers' working status is also positively associated with newborns' weight and, most importantly, educational attainment exhibits a clear positive gradient whereby a greater level of education by mothers is more positively associated with their newborns' birth weight. Relative to infants born to mothers with no education, infants born to mothers with primary, secondary, and higher education weigh 41, 57, and 87 grams more, respectively. Note that the magnitude of the mobile-phone coefficient in this specification is almost identical to the one of primary education–hence, a rather sizeable and meaningful coefficient estimate. Lastly, girls are more likely to have lower weight at birth, in line with the relevant literature [33].

Results on the binary outcome for models 1 and 2 are in line with those discussed above, as the estimated association between mothers' mobile-phone ownership and the probability of their infants being born LBW is negative and strongly significant. In the case of the bivariate association, mothers owning mobile phones are 3.2 percentage points less likely to give birth to LBW infants compared to mothers who do not own mobile phones, corresponding to a 18% decrease in percentage terms. That same estimated coefficient decreases to 2.4 percentage points when individual-level variables are added (corresponding to a 13.5% decrease). Respondents' age is not associated with their infants' probability of being born LBW, while women currently married face the lowest probability of giving birth to low-weight infants, with a negative association of 3.2 percentage points, followed by those "living in a union" (2.4 percentage points), relative to women who have never been married/in a union. Again, the association is negative for women "currently working" and the association between mothers' educational level and their newborns' health exhibits a clear negative gradient whereby the higher the mother's education, the less likely it is that she gives birth to a LBW infant. Similarly, girls have a significantly higher likelihood of being born LBW.

**Table 2. Association between mothers' ownership of mobile phones and infants' birth weight.**

| | (1) | | (2) | | (3) | |
|---|---|---|---|---|---|---|
| | Continuous | LBW | Continuous | LBW | Continuous | LBW |
| Owns a mobile phone | 45.366*** | -0.032*** | 40.841*** | -0.024*** | 31.596*** | -0.018*** |
| | (5.491) | (0.003) | (5.717) | (0.003) | (6.100) | (0.003) |
| Age | | | -0.438 | 0.000 | -0.454 | 0.001 |
| | | | (0.583) | (0.000) | (0.594) | (0.000) |
| Marital status (Ref.: Never in union) | | | | | | |
| Married | | | 67.164*** | -0.032*** | 64.378*** | -0.030*** |
| | | | (11.899) | (0.006) | (11.936) | (0.007) |
| Living with partner | | | 68.109*** | -0.024*** | 69.416*** | -0.023*** |
| | | | (13.486) | (0.007) | (13.561) | (0.007) |
| Widowed | | | 71.137** | -0.026* | 74.878** | -0.026* |
| | | | (29.907) | (0.015) | (30.262) | (0.015) |
| Divorced | | | 49.371** | 0.001 | 47.485** | 0.002 |
| | | | (21.758) | (0.013) | (22.331) | (0.013) |
| No longer living together | | | 78.143*** | -0.011 | 80.410*** | -0.012 |
| | | | (19.854) | (0.010) | (19.988) | (0.010) |
| Currently working | | | 24.011*** | -0.006** | 24.216*** | -0.006** |
| | | | (5.705) | (0.003) | (5.770) | (0.003) |
| Education (Ref.: None) | | | | | | |
| Primary | | | 41.030*** | -0.013*** | 37.302*** | -0.010*** |
| | | | (7.971) | (0.004) | (8.059) | (0.004) |
| Secondary | | | 57.758*** | -0.032*** | 49.319*** | -0.026*** |
| | | | (8.574) | (0.004) | (8.964) | (0.005) |
| Higher | | | 86.033*** | -0.064*** | 69.466*** | -0.054*** |
| | | | (11.863) | (0.006) | (12.666) | (0.007) |
| Sex of the child (Ref.: Male) | | | -104.445*** | 0.034*** | -104.607*** | 0.034*** |
| | | | (4.303) | (0.002) | (4.390) | (0.002) |
| Wealth index (Ref.: Poorest) | | | | | | |
| Poorer | | | | | 25.854*** | -0.007* |
| | | | | | (7.869) | (0.004) |
| Middle | | | | | 41.988*** | -0.013*** |
| | | | | | (8.517) | (0.004) |
| Richer | | | | | 44.655*** | -0.019*** |
| | | | | | (9.842) | (0.005) |
| Richest | | | | | 61.044*** | -0.021*** |
| | | | | | (11.933) | (0.006) |
| Urban residence (Ref.: Rural) | | | | | -29.495*** | 0.006* |
| | | | | | (6.983) | (0.004) |
| Owns TV | | | | | 7.067 | -0.007* |
| | | | | | (7.650) | (0.004) |
| Owns radio | | | | | 2.511 | -0.006** |
| | | | | | (5.589) | (0.003) |
| Constant | 3,294.324*** | 0.127*** | 3,179.542*** | 0.181*** | 3,176.225*** | 0.179*** |
| | (35.728) | (0.016) | (40.872) | (0.019) | (41.408) | (0.020) |
| Birth order dummies | No | No | Yes | Yes | Yes | Yes |
| Country and survey year dummies | Yes | Yes | Yes | Yes | Yes | Yes |
| Observations | 152,172 | 152,172 | 152,158 | 152,158 | 147,630 | 147,630 |

(*Continued*)

**Table 2.** (Continued)

|  | (1) | | (2) | | (3) | |
|---|---|---|---|---|---|---|
| R-squared | 0.032 | 0.018 | 0.045 | 0.026 | 0.046 | 0.026 |

**Notes**: LBW = low birth weight. Standard errors (in parentheses) clustered at the PSU level. DHS sampling weight applied. Country, year, and birth-order dummies omitted from the table. LBW defined as weight below 2,500 grams

*** p<0.01

** p<0.05

* p<0.1.

### Household-level controls

Once household-level controls are added in model 3, the most noticeable change is the lower magnitude of the estimated association between mobile-phone ownership and birth weight, from 41 to 31.6. This most likely suggests that characteristics related to the household are relatively more influential in determining a newborn's weight. Wealth is positively associated with a newborn's weight, distinctly marked by an increasing gradient, whereby the higher the household wealth, the greater the weight of the newborn. Newborns to mothers living in urban settings are likely to weigh 30 grams less compared to those born to mothers living in rural ones. Interestingly, TV and radio ownership are positively associated with newborn's weight, yet they are not statistically significant. Adding household-level variables, moreover, seems to affect the individual-level magnitude of the estimated associations. For example, although the associations are still positive, the coefficient size is reduced for all levels of education, especially at the highest level.

Results on the binary outcome are also in line with those discussed above, yet with some minor differences that suggest, once again, that looking at both outcomes is a sensible strategy. With the addition of the household-level variables, the probability that mothers' mobile-phone ownership affects whether the infant is born LBW is reduced from 2.4 to 1.8 percentage points (10.5% decrease). The influence of wealth is marked by a negative gradient, whereby the higher the amount of wealth in the household, the lower the probability for the mother to give birth to a LBW infant. The association between parents living in urban settings and the probability of their newborn being LBW is positive. Lastly, the association between TV- and radio-ownership and newborn' weight is negative and now statistically significant. Note that when we compare the estimated coefficients on TV and radio with the mobile-phone one, we notice that the magnitude of the latter is close to three times higher than that of the former two, and more strongly predictive of LBW.

### Ancillary analyses

As shown in multiple ancillary analyses reported in the S1 File, these results are very robust to alternative sampling strategies, methodological approaches, and attrition concerns. Moving in order, S3 Table in S1 File provides estimates from the full specification limited to the sample of all births occurring over the previous year. Results are very similar and even stronger: for the continuous outcome, the estimated mobile-phone coefficient is 33.9 (against 31.6), while for the dichotomous outcome, the coefficient is -0.024 (against -0.018). Estimates are virtually identical when limiting the sample to the most recent birth, as shown in S4 Table in S1 File: 31.5 for birth weight (against 31.6) and -0.019 for LBW (against -0.018). Moreover, S5 Table in S1 File provides results from the full specification adding fixed effects at the PSU level (on top of country, year, and birth order), to ensure that there is no community-level heterogeneity or

peculiarity that may be driving the estimates. While results are weaker, especially for the continuous outcome, we keep observing a robust and statistically significant effect size on the mobile-phone coefficient. Conversely, when using two different matching techniques (shown in S6 Table in S1 File), namely nearest-neighbor (*nn*) matching and coarsened exact matching (*cem*), and matching on socioeconomic variables such as respondent's education, household location of residence, wealth index, and country, we find stronger results, with estimated Average Treatment Effects (ATE) ranging between 52.6 and 55 (against 31.6) when predicting birth weight, and -0.029 and -0.032 (against -0.018) when predicting LBW. Nonparametric matching techniques have two distinct features relative to regression-based models: they do not assume any *a priori* functional form for the relationship between mobile phone ownership and infant health outcomes, and they rely on comparing the treatment observations with a closely matched set of control observations, rather than using all the untreated observations in the sample as controls. Moreover, S7 Table in S1 File provides results from the full specification conducting a "bounding exercise" that is useful to test the sensitivity of the estimates to the high number of births with no information on birth weight. Basically, we first assume that all "missing births" are normal weight (*best-case scenario*; very unlikely given that attrition analyses reveal that these are more disadvantaged mothers) and then assume that all missing births are LBW (*worst-case scenario*; more likely). In the former scenario, the estimated coefficient goes to zero (0.001), while in the latter it becomes even higher in magnitude (-0.051 against -0.018), suggesting that results would still hold and would even be magnified. Lastly, as the health and "wealth" relationship is bidirectional, S8 Table in S1 File provides results from the full specification, adding weight of the mother (in kg) as further control. This variable was omitted from the main specification as it is not available in five countries, namely Angola, Indonesia, Philippines, Senegal, and Zambia. These analyses show that mother's health indeed matters, as the coefficient on mobile-phone ownership is cut in half (from 31.6 to 14.5), yet it remains quite relevant.

## Heterogeneity at the individual level

We then explore whether these associations differ by individual- and household-level characteristics, focusing on individual level of education and household location of residence. To do so, we run analyses including an interaction term between mobile-phone ownership and these two measures of socioeconomic status and plot the resulting estimated coefficients. These are provided in Fig 4, which does suggest that associations differ importantly across the educational ladder. Specifically, among women with no education we observe a statistically significant coefficient of 64 (grams) for the birth weight outcome, while this declines to 0.36 (not statistically different from zero) among women with some tertiary education, providing evidence of an overall clear gradient whereby associations are strongest among more disadvantaged women. The same holds for the LBW outcome. Conversely, while we also observe some differences between rural and urban areas–with stronger coefficients in rural areas–these two are not statistically different from each other.

## Potential mediators

Table 3 offers evidence of potential channels that may partly explain the positive association between mobile-phone ownership and birth weight. The table reports estimates from the full specification, adding mediators belonging to the two categories defined previously, namely knowledge and access to information (panel 1), access to healthcare services (panel 2), and all combined (panel 3). Note that for these analyses we are not able to retain the full sample size, as the variables we are using as proxies are not collected in all the countries. As such, in panel 1

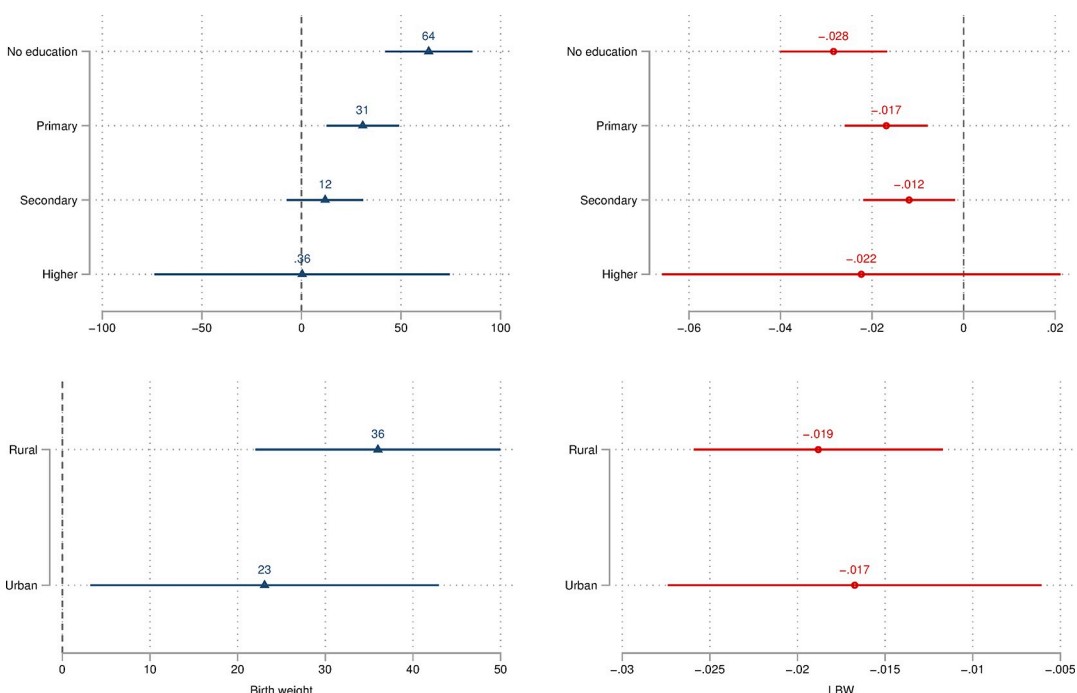

**Fig 4.** Heterogeneity of associations by women's level of education (top panel) and household location of residence (bottom panel). **Notes**: LBW = low birth weight. LBW defined as weight below 2,500 grams. DHS sampling weights applied. 95 percent confidence intervals reported.

we miss Bangladesh, Indonesia, Nigeria, and Tanzania, while in panel 2 we miss Angola, Jordan, Malawi and Nepal. In panel 3, we thus miss all eight countries, bringing down the overall sample to 21 countries. Our results show that knowledge-related channels do not seem to mediate the relationship to a significant extent, except for receiving texts about family-planning, which is associated with a higher birth weight in panel 3. Conversely, we observe stronger associations with mechanisms related to expanded access to healthcare services in both pre- and postnatal periods, as evidenced by the very robust coefficient on the number of antenatal visits. Looking at panel 2, estimates suggest that one additional antenatal visit is associated with higher birth weight by 11.6 grams, a coefficient which gets even bigger (14) when including all mediators together. Similarly, we observe significant negative associations between receiving professional help during birth and the likelihood of having a LBW infant, which decreases by 1.5 percentage points, approximately. All in all, by adding all potential mechanisms, the effect size for mobile-phone ownership gets bigger in magnitude for the continuous outcome (from 31.6 to 34.8), while it is slightly reduced for the binary outcome (from 0.018 to 0.017). While there may be plenty of additional mechanisms at play that we cannot capture with the current information (as also evidenced by the marginal changes in the mobile-phone coefficient), such mediators corroborate the idea that mobile-phone ownership may broaden women's access to maternal-health services by providing swifter and perhaps even remote healthcare support, and by making relevant information more readily available even just by text messages (i.e., no Internet connectivity needed).

## Country-level heterogeneity

We conclude this investigation by running estimates separately by country leveraging country/mobile-phone ownership interactions and plotting the resulting coefficients. S1 Fig in S1 File

**Table 3. Association between mothers' ownership of mobile phones and infants' birth weight, potential channels.**

| | Broadened knowledge and access to health-related information (1) | | Expanded access to healthcare services (2) | | All (3) | |
|---|---|---|---|---|---|---|
| | Continuous | LBW | Continuous | LBW | Continuous | LBW |
| Owns a mobile phone | 29.359*** | -0.019*** | 34.681*** | -0.015*** | 34.854*** | -0.017*** |
| | (7.648) | (0.004) | (6.693) | (0.004) | (8.675) | (0.005) |
| Knows place to get HIV tested | -20.120* | -0.001 | | | -13.895 | -0.005 |
| | (12.087) | (0.006) | | | (14.249) | (0.008) |
| Knows modern method of contraception | -40.468 | -0.022 | | | -26.411 | -0.027 |
| | (37.288) | (0.016) | | | (40.663) | (0.019) |
| Heard family planning by text on phone | 23.268 | 0.001 | | | 36.265* | 0.014 |
| | (15.687) | (0.009) | | | (19.010) | (0.011) |
| Number of antenatal visits | | | 11.586*** | -0.005*** | 14.031*** | -0.004*** |
| | | | (1.309) | (0.001) | (2.133) | (0.001) |
| Birth assisted by health professional | | | -1.332 | -0.010* | 4.192 | -0.015** |
| | | | (9.187) | (0.005) | (12.392) | (0.007) |
| Constant | 3,268.908*** | 0.178*** | 3,057.000*** | 0.253*** | 3,079.281*** | 0.274*** |
| | (58.379) | (0.026) | (29.004) | (0.015) | (51.241) | (0.025) |
| All other individual and HH-level controls | Yes | Yes | Yes | Yes | Yes | Yes |
| Birth order dummies | Yes | Yes | Yes | Yes | Yes | Yes |
| Country and survey year dummies | Yes | Yes | Yes | Yes | Yes | Yes |
| Observations | 101,361 | 101,361 | 85,275 | 85,275 | 54,256 | 54,256 |
| R-squared | 0.046 | 0.022 | 0.050 | 0.030 | 0.049 | 0.025 |

**Notes**: LBW = low birth weight. Standard errors (in parentheses) clustered at the PSU level. DHS sampling weight applied. Country, year, birth-order dummies, and all other controls omitted from the table. LBW defined as weight below 2,500 grams

*** $p < 0.01$

** $p < 0.05$

* $p < 0.1$.

provides such coefficients for the continuous outcome (top panel) and the dichotomous one (bottom panel). Despite some heterogeneity, within the former panel 22 out of 29 coefficients are positive, in line with the evidence provided on the pooled sample. Out of the negative estimates, none of them is statistically significant. Some of the strongest positive associations are observed in Benin, Burundi, Guinea, Nigeria, Nepal, and South Africa. Similar conclusions can be reached by focusing on the bottom panel, as the estimated association is negative in 21 out of 29 countries, in line with the evidence provided on the pooled sample. Again, some of the strongest negative associations can be observed in Burundi, Nigeria, Nepal, Pakistan, and South Africa. These findings provide preliminary suggestive evidence that the decline in the proportion of LBW infants born to mothers with mobile phones is most marked in countries that have some of the highest shares of LBW infants to start with–with two clear examples being Pakistan and Nepal. In these countries, the share of births that are LBW are, respectively, 0.36 (first highest, Fig 1) and 0.29 (third highest, Fig 1), and the associations between ownership of mobile phones and LBW are among the most negative– 4.8 and 9.6 percentage points, respectively.

To test this more formally, we plotted the estimated associations from the full specification predicting LBW for each country against different measures of infant health at the country level. Specifically, we obtained a measure of country-level prevalence of LBW from UNICEF, as well as country-level prevalence of stunting, wasting, and underweight from the DHS

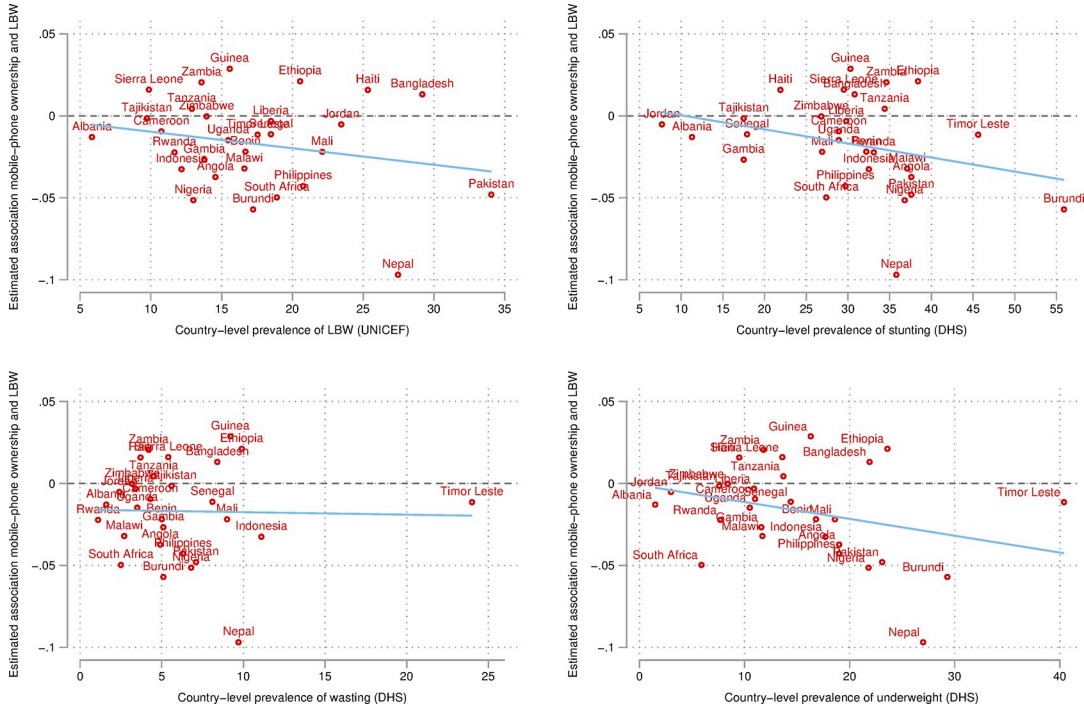

**Fig 5. Heterogeneity of associations by countries' infant health status, predicting LBW.** *Notes*: LBW = low birth weight. LBW defined as weight below 2,500 grams. All estimates are from the same year as the survey. Line provides linear fit (correlation).

StatCompiler. Informed by the Pakistan and Nepal cases, we are interested in assessing whether associations are more (or less) negative in contexts with poorer infant health when looking at all countries together. Fig 5 provides very clear results: associations between mobile-phone ownership and LBW are more negative (i.e., stronger) in countries that have higher prevalence of LBW, stunting, and underweight, while evidence on wasting is weak. The latter finding is also confirmed by ancillary analyses reported in S2 Fig in S1 File plotting interactions (predicted margins) between mobile-phone ownership at the individual level and these same country-level measures of health. Findings are particularly clear for stunting, wasting, and underweight: when country-level health is better, mobile phones do not seem to play a big role (i.e., confidence intervals are overlapping), yet the role of mobile phones becomes stronger and stronger as country-level health gets worse. This finding strengthens the idea that mobile-phone ownership among mothers can be more beneficial for infants' health in countries where infants' health outcomes are less desirable.

Lastly, we conducted a similar exercise focusing on mobile-phone diffusion at the country level (Fig 6). We obtained two measures, one capturing mobile-phone ownership among all women (not just our analytical sample of mothers) from the DHS, and the other capturing mobile-phone subscriptions per 100 people from the World Development Indicators (WDI), originally obtained from the International Telecommunication Union (ITU). While results are less stark, our analyses show that associations are stronger in contexts in which mobile-phone diffusion is higher. Combined with the above finding, this result suggests that investing more heavily in mobile technology and devising policies to lower the cost of data plans may have important benefits for infant health, and for global health and development more broadly.

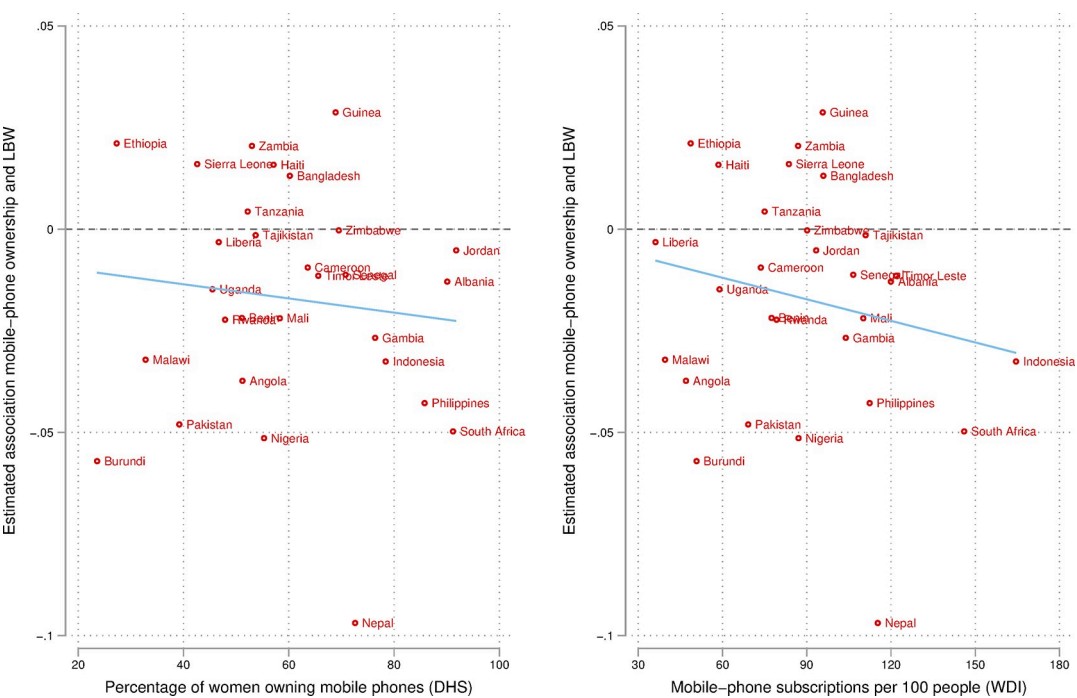

**Fig 6. Heterogeneity of associations by countries' mobile-phone diffusion, predicting LBW.** *Notes*: LBW = low birth weight. LBW defined as weight below 2,500 grams. The DHS estimates pertain to all women in the sample (not to the analytical sample of this study). The WDI estimates pertain to both men and women. All estimates are from the same year as the survey. Line provides linear fit (correlation).

## Conclusions

In this study we have leveraged DHS data from 29 LMICs to explore whether an association exists between mothers' ownership of mobile phones and their children's health at birth. We have found that infants born to women owning mobile phones fare consistently better in terms of birth weight and are less likely to be born LBW, even after accounting for potential socioeconomic confounders such as own level of education, household wealth, or presence of other sources of information in the household. Leveraging information on additional variables that may serve as relevant mechanisms, we have found that this association is explained, albeit to a small extent, by a higher number of antenatal visits, by whether the birth is assisted by professional staff, and by the extent to which mothers hear about family planning by text message, corroborating the idea that mobile phones may enable smoother communication and better community outreach, which in turn allow women to more swiftly connect with other women in their communities, as well as with health professionals/practitioners/facilities, leading to positive infant health outcomes [7, 34]. While the DHS do not provide (yet) information on whether these phones are simple feature phones or smartphones, the channels we highlight are theoretically meaningful irrespective of the enhanced connectivity that smartphones would enable–and the related costs of airtime which are a significant barrier for poor households in LMICs [35]. This is exemplified, for instance, by the family-planning text-message mediator, which relies on simple SMS functions that basic feature phones also possess. In line with previous scholarship on the topic [34, 36], we therefore claim that even simple feature phones may have valuable implications for the lives of women across LMICs, while acknowledging that smartphones–which enable broader connectivity and activate and amplify specific information-seeking channels–would further enhance such potential.

We also found that results are consistent in the vast majority of countries (about 70% of countries), yet important exceptions remain–such as Ethiopia, Liberia, Sierra Leone, and Zambia–which deserve additional investigations, for instance through more detailed information on mobile-phone ownership and usage, which the DHS will release in subsequent rounds. Lastly, we found stronger results in countries where infant health is poorest and mobile-phone diffusion is highest, suggesting the comparatively higher potential of the diffusion of digital technologies in more disadvantaged contexts. Although not identical, such idea resonates with previous findings suggesting that mobile-phone penetration is more beneficial for dynamics of women's empowerment and global development among some of the poorest countries and communities in the world, suggesting diminishing returns as countries' level of development improves alongside, arguably, better infant health [7].

Our results have important implications for global discussions tied to the attainment of the UN SDGs such as better health and wellbeing (mostly maternal and infant health), more gender equality, higher economic growth and, ultimately, reduced poverty. In so doing, our findings complement existing work in rural Malawi focusing on mother's health in the postpartum phase [10] by highlighting benefits and potential channels that mobile phones may enable in terms of broader access to healthcare and reproductive health *before* and *during* pregnancy. While we have no information on digital connectivity nor social media use, such research may also inform recent work assessing the role of behavioral interventions delivered through digital technologies aimed at shaping health outcomes and health equity across different populations and diverse societies [37]–an area of study that is largely missing in the context of LMICs, yet it is highly promising.

This study has limitations that lay the ground for subsequent research. An important one is the cross-sectional nature of the data, which prevents from drawing causal conclusions. Future studies–most likely focused on single-country scenarios–might leverage natural experiments exploiting temporal discontinuity in technology rollout or design randomized controlled trials on mobile-phone expansion to get at the causal nature of the relationship. Following existing research [7, 34, 36, 38], we could have adopted an Instrumental Variable (IV) approach to instrument mobile-phone ownership through variables such as mobile-phone connectivity in the community and/or lightning strikes in the surrounding geographical area–which have been proven to be strong and reliable IVs for mobile-phone ownership–yet we preferred to keep the nature of the current study descriptive, to highlight the crucial importance of careful descriptive work, especially with cross-sectional data. Second, the current variable in the DHS only measures ownership of mobile phones, while existing research suggests that, despite ownership, women may face significant barriers in using phones, including shared device use with their partners [39]. Additional information on intensity of use, digital skills, digital divides, access to electricity, costs of data plans, etc. would greatly enrich the picture and shed more nuanced light on the relationships we documented. We take this shortcoming as a suggestion to policymakers that future data collection efforts should complement information on mobile-phone ownership with additional variables on intensity of use, phone type, quality of phones, and ownership and usage barriers. Third, we identified in this study some mechanisms that may underlie the documented associations. As confirmed by the magnitude of the estimated mobile-phone coefficient, which decreases only minimally when accounting for the aforementioned variables, we caution the reader that there may be multiple other mechanisms–that cannot be tested with the data at hand–that could explain in more detail why we observe the positive associations we document between mobile-phone ownership and infant health. Fourth, we acknowledge that mother's health may be a confounder in the relationship between mobile-phone ownership and infant health, yet the DHS do not provide good information on self-reported health nor complete information for all countries on mothers' current weight.

Lastly, we recall that we have no information on infant health for about 37% of mothers, a set of mothers who are poorer, less educated, and less likely to own a mobile phone. Despite our analytical attempts to overcome this issue, this is a crucial limitation in this study which leads to one important conclusion and one key policy recommendation. The former suggests that the set of "missing mothers" would be the most relevant as they are the ones whose infants have, most likely, poorer infant health, thus suggesting that our estimates may be even strengthened if we had access to birth-weight information for such mothers (in other words, we are under-estimating the "real" association, as demonstrated by the bounding exercise). The latter speaks to scholars and policymakers concerned with devising better strategies to ensure that infants' birth weight is properly collected and recorded across all socioeconomic strata and across all countries and communities in the globe.

## Supporting information

**S1 File.**
(DOCX)

## Acknowledgments

Pesando acknowledges support from the Weatherhead Center for International Affairs at Harvard University and the Division of Social Science at New York University–Abu Dhabi. The authors are grateful for useful comments and suggestions received at the 2023 Population Association of America (PAA) annual meeting, specifically by Drs. Joshua Wilde and Emily Smith-Greenaway.

## Author Contributions

**Conceptualization:** Luca Maria Pesando.

**Data curation:** Komin Qiyomiddin.

**Formal analysis:** Luca Maria Pesando, Komin Qiyomiddin.

**Funding acquisition:** Luca Maria Pesando.

**Methodology:** Luca Maria Pesando, Komin Qiyomiddin.

**Supervision:** Luca Maria Pesando.

**Writing – original draft:** Luca Maria Pesando, Komin Qiyomiddin.

**Writing – review & editing:** Luca Maria Pesando.

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
