## [Decision Letter · Decision Letter 0]

12 Apr 2023

PONE-D-22-35074Digital Technologies and Children’s Health at BirthPLOS ONE

Dear Dr. Pesando,

Thank you for submitting your manuscript to PLOS ONE. After careful consideration, we feel that it has merit but does not fully meet PLOS ONE’s publication criteria as it currently stands. Therefore, we invite you to submit a revised version of the manuscript that addresses the points raised during the review process. Two reviewers have revised the manuscript noting that there are problems that need to be raised. One particular concern is the distance between the title and the actual analysis. Consider changing the title to something more concrete, since the paper is not so much about "digital technologies". Also note the need to provide a better reasoning of the underlying theory.

We look forward to receiving your revised manuscript.

Kind regards,

José Antonio Ortega, Ph.D.

Academic Editor

PLOS ONE

Journal Requirements:

2. Thank you for including your ethics statement:  "N/A".  

a.           For studies reporting research involving human participants, PLOS ONE requires authors to confirm that this specific study was reviewed and approved by an institutional review board (ethics committee) before the study began. Please provide the specific name of the ethics committee/IRB that approved your study, or explain why you did not seek approval in this case.

b.           Please provide additional details regarding participant consent. In the ethics statement in the Methods and online submission information, please ensure that you have specified (1) whether consent was informed and (2) what type you obtained (for instance, written or verbal, and if verbal, how it was documented and witnessed). If your study included minors, state whether you obtained consent from parents or guardians. If the need for consent was waived by the ethics committee, please include this information.

Reviewers' comments:

Reviewer's Responses to Questions

**Comments to the Author**

1. Is the manuscript technically sound, and do the data support the conclusions?

Reviewer #1: Yes

Reviewer #2: Yes

2. Has the statistical analysis been performed appropriately and rigorously? 

Reviewer #1: Yes

Reviewer #2: No

3. Have the authors made all data underlying the findings in their manuscript fully available?

Reviewer #1: Yes

Reviewer #2: Yes

4. Is the manuscript presented in an intelligible fashion and written in standard English?

Reviewer #1: Yes

Reviewer #2: Yes

5. Review Comments to the Author

Reviewer #1: Thank you for the opportunity to review the manuscript. It is incredibly insightful, analyses are very well done and easy for any reader to follow. Bravo! A few comments for consideration

Abstract:

1) please elaborate on what " shape global development outcomes" means...could be more specific in stating that it shapes the SDG outcomes and reference the WHO's digital health agenda to strengthen the statement

2) POWERFUL! "Partly, this association

is mediated by a higher number of antenatal visits and by the extent to which mothers

hear about family planning by text message."

3) Also VERY powerful! "Results are stronger in countries where

infant health is poorest, suggesting the comparatively higher potential of the diffusion of

digital technologies for global development in poorest contexts."

Intro:

1)see comment #1 of abstract critique. Done BEAUTIFULLY in abstract

2) The field is slowly moving away from the term “sub Saharan Africa” due to its negative connotation and instead using “Africa south of the Sahara” or “The African region” or if need be, “resource-limited settings of the African region “ as not all of the region is as “poor” as is often depicted.

Please see resources below:

"The term spread as a replacement for the racially-tinged phrases “Tropical Africa” and “Black Africa” that were used until around the 1950s, says Columbia University anthropologist Brian Larkin."

http://www.cnn.com/2010/WORLD/africa/09/20/sub.saharan.africa/index.html

https://qz.com/africa/770350/why-do-we-still-say-subsaharan-africa

3)"little research to date has focused on the

intergenerational implications of digital technologies for child health and wellbeing"

why is an intergenerational focus important? please elaborate a little more

4) similarly, why is the infant's birth weight (of all the metrics available for infant health), important as a relevant outcome for the field? What does birth weight predict for child development and other relevant outcomes?

"– bears any relationship with their offspring

health at birth as measured, for instance, by birth weight."

and this

"no

study has focused explicitly on the relationship between mothers’ mobile-phone ownership and

infants’ health as measured by birth weight and low birth weight (LBW) as we do in this study."

LBW is selected randomly it seems, and must be justified by the authors in its selection.

5) Beautiful section "EXISTING EVIDENCE FROM LMICs ON THE ROLE OF MOBILE PHONES IN

THE HEALTH ARENA"

6) i'd remove the "now" in this sentence "mothers can now

call a helpline to inquire and ask questions about their child’s health conditions"

7)Not sure why the "so called" is necessary in these phrases? "highlighting again access (so-called, first-level) and skill-related (so-called, second-level) digital divides" Also those need references.

Methods

8) why were Armenia and Maldives excluded from the

final sample selection? What was the reason?

9) beautiful discussion of reverse causality

10) great discussion of sampling bias with the "advantaged" mothers

11) the authors have no knowledge of this phrase is true or not, so I'd remove

" (twins, most likely)."

12) the methods are nice and clear and easy to follow. for example this is BEAUTIFUL

"In other words, over the previous three years, 170,916 mothers had 260,158 births,

of which only 152,172 have complete birth-weight information"

13) this phrase does not make sense to the reviewer. how does 2% difference range translate to 6% and 25% difference? Also, 2% is not a statistically significant difference...whether raw or adjusted...perhaps the solution is to translate this 2% difference to population level data. 2% represented XYZ number of babies in Bangladesh and XYZ number of babies Albania...is that what the authors are trying to say? Would make more sense to the reader if so.

"As the variability in LBW is high across countries,

a 2 percentage-point difference may imply percentage changes ranging between 6% in Bangladesh

to 25% in Albania."

14) Table 1: define LBW cutoffs in tables for the reader

15) A figure will help the reader fully grasp the directionality of the potential mediators and their effect sizes

16) Please speak to this a little more by walking the reader through the data for either country as it's a mind-twist:

"What is even more important is that the decline in the proportion of LBW

infants born to mothers with mobile phones is most marked in countries that have some of the

highest probability of LBW infants born to mothers without mobile phones – with two clear

examples being Pakistan and Bangladesh. "

Discussion

17) This phrase needs a reference

"corroborating the idea that mobile phones enable smoother communication and better

community outreach, which in turn allow women to more swiftly connect with other women in

their communities, as well as with health professionals/practitioners/facilities, leading to positive

infant health outcomes."

18) The discussion could benefit from some discussion on the importance of the results with regards to child/human development, SDGs, maternal health, access, etc etc etc. Just re-iterating the results is not sufficient. Please see :

1) Anto-Ocrah, M., Latulipe, R.J., Mark, T.E. et al. Exploring association of mobile phone access with positive health outcomes and behaviors amongst post-partum mothers in rural Malawi. BMC Pregnancy Childbirth 22, 485 (2022). https://doi.org/10.1186/s12884-022-04782-0

2) Healthy People.gov. Social Determinants of Health. 2014 2019 3/20/2019]; Available from: https://www.healthypeople.gov/2020/topics-objectives/topic/social-determinants-of-health.

3)Petkovic J, Duench S, Trawin J, Dewidar O, Pardo Pardo J, Simeon R, et al. Behavioural interventions delivered through interactive social media for health behaviour change, health outcomes, and health equity in the adult population. Cochrane Database Syst Rev. 2021;5:Cd012932.

19)the figures are great but pixelated

Reviewer #2: This study examines the cross-sectional relationship between mobile phone ownership and birthweight. It is an interesting topic; however, I have several comments about the framework/theory of change, methods, and presentation of results:

Introduction

1. On page 6, the authors write “Despite the obvious connection between most of these interventions and children’s health outcomes, alongside the myriad opportunities enabled by mobile phones in the health arena, no study has focused explicitly on the relationship between mothers’ mobile-phone ownership and infants’ health as measured by birth weight”. Unfortunately, I do not find this particularly obvious. (Moreover this sentence comes after a whole paragraph on barriers to using mHealth so it’s not clear what “these interventions” even refers to.) In general, the introduction would benefit from a stronger link between what is known about mobile phones and maternal/reproductive health and how that would theoretically impact birthweight. The authors list a variety of studies in very different areas – HIV, measles, contraception, etc but it’s not clear how any of these mobile phone programs would impact birth weight. I suggest the authors focus on the relevant studies and country mHealth programs (e.g. measles vaccination is irrelevant as the child would already be born), as well as provide a framework or model to indicate the mechanisms through which those interventions could change birthweight.

Methods –

2. I agree with the authors that missing birthweight for 36% of the sample is a significant issue. I recommend the use of multiple imputation to impute low birth weight status from the predictor variables. Another alternative is to do a sensitivity analysis in which 100% of those missing are assumed LBW (as a worst case scenario) and see if results stand up. These observations are clearly not missing at random based on the Table A1, so the analysis should not treat them that way.

3. The authors include multiple births (siblings or twins) from individual women. This results in clustering in the data at the woman-level. As the exposure is woman-level, I would recommend conducting the analysis that way. For example, the authors could take the most recent birth, or a birth at random for each woman, rather than including all births.

4. The mediator analysis needs more careful attention. First, there is no justification or motivation for the use of these mediator variables (what are the theoretical pathways through which mobile phones could improve birth weight?) This should also be included in the introduction as part of the theoretical framework or model.

5. Second, the first three mediator variables are likely very highly correlated. The number of antenatal visits is not very different from whether the mother saw a health professional for antenatal care. The same goes for visiting a health facility in the previous 12 months, which they would do if they were getting antenatal care. I would not include all three of these into the same model as they are capturing the same thing – access to health care, which is only one pathway. The authors should choose one of these three to capture health care access. The last mediator on hearing family planning text messages is good, though as I mention in point 4 above, it needs to be justified as to what the authors think it is measuring in terms of the mediator pathway. What about other pathways, for example, are there any knowledge variables in the DHS that the authors could explore with regards to family planning/child health? I would imagine that access to a mobile phone could improve knowledge of health. There may be other pathways as well.

Results

6. It would be helpful to have a table of descriptive characteristics of women for the exposure, all controls and all mediator variables.

7. There is likely residual confounding by SES, despite controlling for education and wealth index. Perhaps the authors could include an asset index. I see the controls for owns TV and radio, but I believe the DHS includes a list of assets that can be included as an index. That could reduce some of the residual confounding by SES.

8. I would also include mother’s health variables, including weight, self-reported health status, and any other health indicators that are available, as those could be confounders (health and wealth relationship is bidirectional).

9. Figure 4 is not interpretable. It’s impossible to see anything in these two figures. The authors could change the margins plot to show a bar plot perhaps, by country. Or else present this information in a table with adjusted prevalences and CIs, which may make more sense than a figure if you are trying to show small differences by country.

10. In addition for Figure 4, I would remove the antenatal visit variable as it is on the causal pathway and as far as I understand, the authors are trying to estimate the adjusted difference in risk of low birthweight by phone status. Including variables on the causal pathway biases these estimates.

Discussion

11. The authors wrote “Lastly, we found stronger results in countries where infant health is poorest, suggesting the comparatively higher potential of the diffusion of digital technologies in more disadvantaged contexts.” I don’t see the analysis that shows poorer countries benefit more? Is this just eyeballing it from Figure A1? If that is the case, the authors should conduct this analysis formally, for example, by including country GDP as an interaction variable. This was also repeated in the abstract.

12. It would also be interesting to examine patterns of country-specific results by other variables. For example, do countries with higher mobile phone ownership have stronger results? Or more urban populations?

Minor points

The analysis should control for infant sex as there is notable gender difference in birthweight and LBW.

Please include in the appendix the year of the DHS survey included for each country so that others may be able to replicate the results.

6. PLOS authors have the option to publish the peer review history of their article (what does this mean?). If published, this will include your full peer review and any attached files.

Reviewer #1: No

Reviewer #2: No

---

## [Decision Letter · Decision Letter 1]

25 May 2023

PONE-D-22-35074R1Mobile Phones and Infant Health at BirthPLOS ONE

Dear Dr. Pesando,

Thank you for submitting your manuscript to PLOS ONE. After careful consideration, we feel that it has merit but does not fully meet PLOS ONE’s publication criteria as it currently stands. Therefore, we invite you to submit a revised version of the manuscript that addresses the points raised during the review process. The manuscripts has improved much from the previous round including the changes introduced by the authors following discussion in academic conferences  Among the 2 previous referees, referee 1 was not available but their comments seem to have been addressed. Referee 2 recommends acceptance, but in reality provides comments to include in the manuscript with which the editor aggrees, in particular the robustness check that can be inluded as an appendix and a note on the limitations section. If this is carried out, it will no be necessary to send the manuscript back to the reviewers.

We look forward to receiving your revised manuscript.

Kind regards,

José Antonio Ortega, Ph.D.

Academic Editor

PLOS ONE

Journal Requirements:

Reviewers' comments:

Reviewer's Responses to Questions

**Comments to the Author**

1. If the authors have adequately addressed your comments raised in a previous round of review and you feel that this manuscript is now acceptable for publication, you may indicate that here to bypass the “Comments to the Author” section, enter your conflict of interest statement in the “Confidential to Editor” section, and submit your "Accept" recommendation.

Reviewer #2: All comments have been addressed

2. Is the manuscript technically sound, and do the data support the conclusions?

Reviewer #2: Yes

3. Has the statistical analysis been performed appropriately and rigorously? 

Reviewer #2: Yes

4. Have the authors made all data underlying the findings in their manuscript fully available?

Reviewer #2: Yes

5. Is the manuscript presented in an intelligible fashion and written in standard English?

Reviewer #2: Yes

6. Review Comments to the Author

Reviewer #2: The authors have addressed all of my concerns very well. The theoretical background and motivation for the mechanism analysis has been much improved and strengthened, and I appreciate the series of robustness checks that have been done. The new analyses in figures 4-6 much improve the paper. I have a couple of remaining points:

1. Page 7: “given that birth weight mostly reflects choices, behaviors, and constraints faced by mothers pre- and during pregnancy”

-While behavior is certainly an important determinant, I think that sentence is too strong. There are highly predictive maternal non-modifiable characteristics that affect birth weight including mother’s own birth weight, mother’s height, mother’s age, parity, etc.

2. The robustness check that includes mother’s weight as a control is important to highlight, given the large reduction in point estimate for birthweight and LBW. However, I do not see this robustness check referenced in the text or in the limitations as the authors stated. Please include this analysis in the appendix and add the text to the robustness checks paragraph and to the limitations discussion.

7. PLOS authors have the option to publish the peer review history of their article (what does this mean?). If published, this will include your full peer review and any attached files.

Reviewer #2: No

---

## [Editor Report · Decision Letter 2]

19 Jun 2023

Mobile Phones and Infant Health at Birth

PONE-D-22-35074R2

Dear Dr. Pesando,

We’re pleased to inform you that your manuscript has been judged scientifically suitable for publication and will be formally accepted for publication once it meets all outstanding technical requirements.

Kind regards,

José Antonio Ortega, Ph.D.

Academic Editor

PLOS ONE

Additional Editor Comments (optional):

The changes in the last version addressed the remaining issues satisfactorily.
---

## [Editor Report · Acceptance letter]

26 Jun 2023

PONE-D-22-35074R2 

Mobile Phones and Infant Health at Birth 

Dear Dr. Pesando:

I'm pleased to inform you that your manuscript has been deemed suitable for publication in PLOS ONE. Congratulations! Your manuscript is now with our production department. 

Kind regards, 

on behalf of

Dr. José Antonio Ortega 

Academic Editor

PLOS ONE